# The Role of Neutrophil Extracellular Traps in the Outcome of Malignant Epitheliomas: Significance of CA215 Involvement

**DOI:** 10.3390/diagnostics14030328

**Published:** 2024-02-02

**Authors:** Mihai Emanuel Himcinschi, Valentina Uscatescu, Georgiana Gherghe, Irina Stoian, Adelina Vlad, Delia Codruța Popa, Daniel Coriu, Andrei Anghel

**Affiliations:** 1Department of Biochemistry and Pharmacology, Discipline of Biochemistry, “Victor Babes” University of Medicine and Pharmacy, 300041 Timisoara, Romania; mhimcinschi@gmail.com (M.E.H.); biochim@umft.ro (A.A.); 2Department of Hematology, Fundeni Clinical Institute, 022328 Bucharest, Romaniadaniel_coriu@yahoo.com (D.C.); 3Department of Functional Sciences I/Biochemistry, Faculty of Medicine, Carol Davila University of Medicine and Pharmacy, 050474 Bucharest, Romania; irina.stoian@umfcd.ro; 4Department of Functional Sciences I/Physiology, Faculty of Medicine, Carol Davila University of Medicine and Pharmacy, 050474 Bucharest, Romania

**Keywords:** neutrophil extracellular traps (NETs), NETosis, cancer therapy resistance, epithelioma, IgG, CA215

## Abstract

Neutrophil extracellular traps (NETs) were originally discovered as a part of the innate immune response of the host to bacteria. They form a web-like structure that can immobilize microorganisms or exhibit direct antimicrobial properties, such as releasing reactive oxygen species (ROS). NETs are established when neutrophils undergo a sort of cellular death following exposure to ROS, chemokines, cytokines, or other soluble factors. This process results in the release of the neutrophil’s DNA in a web-like form, which is decorated with citrullinated histones (H3/H4-cit), neutrophil elastase (NE), and myeloperoxidase (MPO). Emerging studies have put into perspective that NETs play an important role in oncology as they were shown to influence tumor growth, malignant initiation, and proliferation, mediate the transition from endothelial to mesenchymal tissue, stimulate angiogenesis or metastasis, and can even help cancer cells evade the immune response. The role of NETs in cancer therapy resides in their ability to form and act as a mechanical barrier that will provide the primary tumor with a reduced response to irradiation or pharmaceutical penetration. Subsequently, cancer cells are shown to internalize NETs and use them as a strong antioxidant when pharmaceutical treatment is administered. In this review, we explored the role of NETs as part of the tumor microenvironment (TME), in the context of malignant epitheliomas, which are capable of an autonomous production of CA215, a subvariant of IgG, and part of the carcinoembryonic antigen (CEA) superfamily. Studies have shown that CA215 has a functional Fc subdivision able to activate the Fc-gamma-RS receptor on the surface of neutrophils. This activation may afterward stimulate the production of NETs, thus indicating CA215 as a potential factor in cancer therapy surveillance.

## 1. Introduction

Neutrophils are the most abundant white blood cells in our immune system and play a crucial role in general inflammation. As for fighting infection, they use various mechanisms to combat pathogens, including phagocytosis, releasing reactive oxygen species, and discharging mediators that can regulate microbial interactions with other cells. Recently, scientists have discovered that neutrophils can also use their intracellular DNA as a defense mechanism, a process known as NETosis [1]. The end product of this phenomenon, called Neutrophil Extracellular Traps, is considered to be a web-like structure that binds molecules like histones (H3, H4), neutrophil elastase, or myeloperoxidase.

NETosis initiates with neutrophil activation and proceeds through the lytic/suicidal phase, where most NETs-generating processes occur. The main mechanism underlining this phenomenon involves the NADPH oxidase 2 (NOX2) enzymatic complexes, used by the cell for their capacity of electron transfer across the biological membranes [2]. NOX2, also known as gp91phox, is expressed primarily in phagocytes (macrophages, neutrophils, and dendritic cells) and is responsible for generating reactive oxygen species (ROS) after the cell’s contact with different stimuli such as pathogens, cytokines, and different mediators of inflammation, via specific receptors, including Toll-like receptors (TLRs), NOD-like receptors (NLRs), cytokine receptors and Fc receptors [2,3,4]. Later on, the ROS generated previously will interact with a large variety of small molecules such as hydrates of carbon, nucleic acids, lipids, and proteins, thus making the neutrophil capable of liberating the content of azurophilic granules, rich in myeloperoxidase (MPO) and neutrophil elastase (NE) [4]. The presence of the last two molecules can influence the structural integrity of the histones, eventually resulting in the rupture of the cell with adjacent loosening of the space between euchromatin and heterochromatin, with the formation of a homogenous intranuclear content [5].

## 2. Tumor Microenvironment

A tumor microenvironment (TME) encompasses all the components, cellular or noncellular, that have a direct or indirect interaction with the tumor and its surroundings. This ecosystem, including cancer cells, immune cells, extracellular matrix proteins, and growth factors, is remarkably dynamic [6]. It plays a crucial role in determining the cancer response to therapy, the primary tumor’s ability to migrate through metastasis, and the development of the primary tumor itself [7,8]. The TME’s heterogeneity is marked by distinct cellular profiles in each region, each functioning as an individual entity. Depending on the development phase, this entity can either provide a pro-cancerous or anti-cancerous substrate. Histologically, the TME can be classified into two main categories: the stroma, which includes adhesion proteins of the extracellular matrix (ECM), newly formed blood vessels, and fibroblasts, and the immune infiltrate, which comprises white blood cells such as neutrophils, B and T cells, macrophages, and natural killer cells [9,10]. The ECM can influence cancer cells’ proliferation, invasion, and progression through proteins like fibronectin and collagen, regulating signaling pathways [9]. The other component, the immune infiltrate, succeeds in conferring the TME a chronic pro-inflammatory status that will further stimulate growth and tumoral expansion. Cancer cells secrete factors that suppress fundamental immune functions, including T cell activation and natural killer cell cytotoxicity. They act by recruiting various myeloid-derived suppression immature cells or regulatory T cells from the hematopoietic tissue. This function is most relevant in the early stages of primary tumor growth and development and is used by cancer cells to protect themselves against the immune system [9,10].

## 3. NETs as Part of the TME

As shown earlier, the immune infiltrate plays an important role in tumor development and progression [11,12]. The literature describes two major categories of mechanisms involved in the formation of NETs: direct and indirect. In the direct mechanism, the tumor is responsible for inducing NETosis by releasing certain molecules such as interleukins or granulocyte colony-stimulating factors (G-CSF) [13,14,15,16,17]. On the other hand, in the indirect one, the pro-inflammatory status of the TME or the apoptosis observed in the early stages of tumorigenesis activates the neutrophils [9,18,19]. G-CSF as well as interleukins can stimulate the production of NETs by activating the NADPH oxidase complex [20]. These molecules are used to further attract neutrophils which can then trigger a chain reaction [18]. Research has shown that IL-1, IL-6, IL-8, IFN, TNF-α, C3a, CXCL1, and other factors play a crucial role in activating specific pathways in NETosis expression, leading to inflammation [21,22,23,24]. NETs are also able to promote T cell exhaustion in the tumor microenvironment [25], or suppress the tumoricidal effect of natural killer cells [26].

## 4. The Roles of NETs in Cancers

To gain a better understanding of how NETs, a complex and multifactorial phenomenon, can affect the development of different types of tumors, researchers have divided this area of study into two main directions: a pro-tumorigenic perspective and an anti-tumorigenic one. In this review, we will focus on the former category and provide examples.

### 4.1. NETs Facilitate the Endothelial-to-Mesenchymal Transition (EndMT)

In the process of acquiring a mesenchymal phenotype, the endothelial cells undergo transdifferentiation, losing their specific membrane markers [27]. As described by Pieterse et al. [28], the mechanism involves neutrophil elastase acting as a proteolytic enzyme for VE-cadherin, an elastase-specific substrate, and recurrent activation of the β-catenin signaling pathway. The β-catenin protein is found in close contact with the cytoplasmic portion of VE-cadherin in all endothelial cells [29]. Its role is to regulate the adhesion between cells and the process of gene transcription necessary for the cells to acquire their mesenchymal phenotype [30]. Therefore, elastase promotes the loss of intercellular connections (cadherin-mediated), leading to the release of β-catenin and activation of its signaling pathway, resulting in EndMT [28].

### 4.2. NETs Are Shown to Have a Positive Effect on Tumor Progression, Invasion, and Growth

Recent studies have established a strong link between tumor growth and NET production. Demers et al. [31] compared tumor growth in peptidyl arginine deiminase 4 (PAD4)-deficient mice to healthy (wild-type) specimens. PAD4 is a critical enzyme that stimulates NETosis by deaminating the arginine on H3 and H4 histones, leading to chromatin release from the neutrophil. Following inoculation of both experimental groups with a culture of Lewis lung carcinoma (LLC), a 35% reduction in tumor volume was reported in the study group versus the control. Proposed mechanism for NET-induced metastasis and tumor growth showed in Figure 1.

A similar experiment conducted by Miller-Ocuin et al. [32] provided that comparable outcomes apply to pancreatic cancer. The median survival rate for wild-type mice was substantially lower (41 days) than the one recorded in the case of PAD4-deficient mice (118 days). Additionally, this second study provided key correlations between circulating DNA in the form of MPO-DNA (a marker widely used to determine a quantitative evaluation of NETosis) and the clinical stages of patients with pancreatic adenocarcinoma. The authors suggested that the serum levels of MPO-DNA might be used as a relevant prognostic value, with higher concentrations being detrimental to the patient.

NETs’ pro-metastatic role is primarily understood from a mechanical viewpoint [33], based on the observation that when neutrophils are activated within blood vessels, they can produce NETs in various organs. NETs then act as a physical barrier, trapping cancer cells that have become detached from the primary tumor [34].

More recently, Yang et al. [35] made a significant contribution that further closed the gap between the production of NETs and metastasis. They initially examined liver biopsy samples collected from 544 patients diagnosed with breast cancer. The metastatic lesions had a high level of the MPO-DNA complex, which was also found in elevated amounts in their serum. This observation indicated MPO-DNA as a potential marker for predicting the metastasis development. In the same study, the timing of NETs formation concerning the increase in seric levels of MPO-DNA and hepatic metastases was investigated. By using immunocompromised mice with MDA-MB-231 cells implanted in the mammary glands, it was reported that NETosis begins much earlier in the pre-metastatic organ (day 16) than the detection of the circulating complex (day 34) or the evidence of liver metastasis. In addition, a significantly lower incidence of metastatic events was observed in a group of PAD4-deficient mice when compared to the control.

### 4.3. NETs Have a Positive Angiogenic Effect

Polymorphonuclear neutrophils represent the main cell type mobilized in the acute phase of inflammation. More importantly, via Toll-like receptors or Fc receptors, the process of NETosis can occur, mediated by the MPO-ROS signaling pathway. This process provides valid evidence that the cell’s pro-angiogenic effect is one of the many side phenomena that occur simultaneously with the appearance of NETs [36,37,38,39].

In a study published by Webb et al. [40], it was shown that activated neutrophils can overexpress the gene responsible for producing vascular endothelial growth factor (VEGF), also known as a vascular permeability factor and promoter of mitosis for the endothelial cells [41]. The article compared the gene expression correlated with the production of VEGF between activated and non-activated neutrophils in vitro, using real-time chain polymerization. The results were significant in favor of the first category. As the cultured cells are stimulated, it is revealed that the majority of the VEGF is produced in the first hour and continues for as long as 4 h. This finding might indicate that there is a concomitant connection between NETs and the concentration of the growth factor.

Similarly, Scapini et al. [42] demonstrated that VEGF mRNA is overexpressed in neutrophils when stimulated with TNF-alpha. The study concluded that the extracellular concentration of VEGF is higher in the artificially stimulated cell culture compared to the unstimulated one.

## 5. NETs Can Severely Affect the Outcome of Cancer Therapy

Resistance to cancer therapy is an important prognostic factor that influences the survival rates of patients. As neutrophil activation and recruitment are present in most solid tumors, it is important to establish if and how the presence of NETs in the TME might influence the outcome of cancer therapy [43]. In the past, low levels of circulating neutrophils were associated with higher survival rates for patients who underwent different cancer treatments, which was initially considered coincidental [44,45].

### 5.1. NETs Can Provide Resistance to Chemotherapy

Clinical studies reveal a significant association between high NET serum levels and reduced chemotherapy efficacy [12]. For example, Ramachandran et al. [46] reported a chemoprotective role of NETs in multiple myeloma patients treated with doxorubicin. By using confocal microscopy and flow cytometry, the study proposed a mechanism whereby cancer cells could internalize the web-like structure of NETs, which acts as a strong antioxidant and binds with doxorubicin [47]. The researchers found that in MM-bearing mice, myeloid-derived suppressor cells (MDSCs), especially polymorphonuclear MDSCs (PMN-MDSCs), were significantly more prevalent in the bone marrow compared to tumor-free mice. Intriguingly, both MDSCs from MM-bearing mice and similar cells from tumor-free mice notably diminished the effectiveness of chemotherapy agents, doxorubicin and melphalan, on mouse MM cell lines. This suggests a protective role of these cells against chemotherapy-induced cytotoxicity. Further, the study extended these findings to human cells, demonstrating that both PMN-MDSCs and mature neutrophils from the bone marrow of MM patients significantly reduced the cytotoxic effects of these chemotherapy drugs on human MM cell lines. The protective mechanism of these cells was found to be distinct from other bone marrow cells, relying on soluble factors rather than direct cell contact. This indicates a unique protective pathway that may involve a range of cytokines and growth factors known to modulate tumor cell chemosensitivity. These findings have significant implications for cancer treatment, highlighting the potential benefit of targeting MDSCs to enhance the efficacy of combined chemo- and immunotherapy treatments [46]. Notably, when the web-like structure is dissolved using DNase, a potent cytotoxic effect is restored.

### 5.2. NETs Can Provide Resistance to Immunotherapy

Several molecules have been identified and used in the last decade to promote the anti-cancer activity of certain immune cells. Immunotherapy focuses on the endogenous immune capacities of the host, which increase dramatically after inhibiting checkpoint molecules, such as PD-L1 and PD-1, known to exert a physiological role in preventing auto-immune responses. Cancer cells use these molecules to escape the immune response and continue to develop [48]. However, an amplification in the recruitment of neutrophils, objectified by the total count in the bloodstream, is associated with a significant decrease in the effectiveness of checkpoint blockade immunotherapies [49].

Zhang et al. [50] recently demonstrated that neutrophil extracellular traps derived from interleukin-17-activated neutrophils can mediate checkpoint blockade immunotherapy in pancreatic cancer. The study found that there was an elevated production of IL-17 in mice and patients with pancreatic cancer, while IL-17 is known for playing a crucial role in the progression and initiation of premalignant pancreatic lesions. This production led to the recruitment of neutrophils and an upregulation of PD-1 and PD-L1 expression in CD8+ T-cells [50]. The finding was further validated by comparing the results with a second study group that received pharmacological inhibitors of IL-17 signaling pathway, who have shown an increase in CD8+ activity, as well as a reduction in their circulating blood count. IL17 plays a critical role in modulating the tumor microenvironment, particularly influencing the recruitment and function of neutrophils. IL17 neutralization was shown to reduce myeloid cell recruitment and increase activation and exhaustion markers in CD8+ T cells. This remodeling of the pancreatic tumor microenvironment by IL17/IL17R signaling affects the spatial distribution and activation of CD8+ T cells, favoring their exclusion and inactivation in the tumor. Furthermore, the study explores the pharmacological and genetic blockade of IL17 signaling as a method to overcome resistance to immune checkpoint inhibition [50]. Despite the IL17 blockade’s positive immunomodulatory effects, single-agent therapy did not yield significant antitumor efficacy. However, a synergistic antitumoral effect was observed when the IL17 blockade was combined with PD-1 inhibition. This combination was effective in different preclinical models of PDAC and showed dependency on CD8+ T cell activation. The study also identifies metabolic changes, particularly lactate levels, as potential biomarkers for the activity of IL17 and PD-1 blockade combination therapy. Moreover, it was demonstrated that IL17 enhances its immunosuppressive effects by promoting neutrophil infiltration and NETosis in pancreatic tumors. The blockade of neutrophils or Padi4-dependent NETosis, in combination with PD-1 inhibition, led to a significant reduction in tumor growth. These findings indicate that IL17 plays a pivotal role in PDAC immunosuppression and resistance to immune checkpoint blockade through its effects on neutrophils and NETosis [50].

Similarly, a study conducted by Alvaro Teijeira et al. [51] proposed that neutrophil extracellular traps create a physical barrier able to impede the contact between cancer cells and T-cells or natural killer cells, thereby reducing the cytotoxic effect of immunotherapy. By using DNase, NETs were removed from the surface of cancer cells and the effector-target contact between them and the immune cells was restored. A proposed role of how NETs can influence cancer therapy is showed in Figure 2.

### 5.3. NETs Can Provide Resistance to Radiotherapy

Radiotherapy, either alone or in combination with other treatments, is a common curative approach for patients with various types of cancers, and resistance to it is a major obstacle in improving oncologic treatment outcomes. Consequently, increasing efforts have been focused on understanding the mechanisms that cause cancer to become resistant to radiation [52,53,54].

When studying the tissue’s normal response to irradiation, an initial fast influx of neutrophils can be observed. This is a primary response aimed at reducing inflammation. However, some studies suggest that tissues may develop resistance to radiotherapy as the immune infiltrate of neutrophils comes into direct contact with soluble factors that can stimulate NETosis [53].

A group of researchers led by Shinde-Jadhav conducted experiments on mice with invasive bladder cancer to understand how the production of NETs can affect cancer’s resistance to radiation [55]. They proposed a mechanism where NETs, induced by irradiation or other factors dependent on TME, can coat the surface of cancer cells, acting as a mechanical barrier and lowering the efficiency of the treatment [55]. The study showed a significant increase in NETs in irradiated tumors compared to non-irradiated ones, leading to a diminishing radiotherapy sensitivity over time. Clinically, the relevance of these findings was evaluated in a cohort of human MIBC patients treated with RT. The study found that NETs were present in the tumor immune microenvironment (TIME) of these patients, particularly in those who did not respond to RT. A high ratio of intratumoral PMNs to CD8 T-cells, which correlates with the presence of NETs, was associated with poorer overall survival. These findings suggest that NETs can impede RT effectiveness by hindering intratumoral CD8 T-cell infiltration, thereby promoting tumor radioresistance [55]. However, when DNase was added to the therapeutic protocol, the response to irradiation was dramatically restored [55]. It was reported also that a protein called high mobility group box protein-1 (HMGB1), which is produced excessively in several types of cancers and acts as a trigger of inflammation, has a selective affinity for the Toll-like receptors known to induce NETosis when stimulated. Thus, it can influence radiotherapy resistance by increasing NET production [55,56,57]. The most important ways that NETs can influence cancer regulation or cancer therapies are summarized in Table 1.

## 6. CA215 as an Immunoglobulin G Variant

Cancer antigen 215 (CA215) is a widely recognized pan-cancer marker in the literature. It is a subtype of immunoglobulin G (IgG) that belongs to the immunoglobulin superfamily (IgSF) and is produced by cancer cells [58,59]. It differs from the usual IgG by presenting a unique carbohydrate-associated epitope, but its Fc region is almost identical to the active region of a normal IgG1. Thus, as part of the IgSF, CA215 may be able to use its intact Fc region to bind to the Fc neutrophil receptors and may be able to initiate NETosis [60,61]. Lee et al. [62] reported that the carbohydrate epitope has a different composition from a typical IgG or the monoclonal antibody RP215, used to measure the levels of CA215 in the bloodstream. The carbohydrate epitope has a lower percentage of N-acetylglucosamine and a higher percentage of mannose, which ultimately gives the molecule its affinity for RP215.

## 7. Cancer Cells and CA215 Production

Over the years, several articles have noted that cancer cells, specifically epithelial cancer cells like thyroid cancer [63], prostate cancer [64], pancreatic cancer [65], breast cancer [66,67], and others, produce a high level of immunoglobulin. These cells also have a high level of glycans, which contribute to their development and growth [68]. It has been suggested that some glycoconjugated molecules, that can maintain their main effector functions (subdivision Fc) like a normal IgG and can also be recognized by RP215 by binding the same carbohydrate [69,70], such as CA215 or CA19-9, might be involved in positively modulating the overall progression of the tumor.

Different types of epitheliomas have been observed to independently mediate the autonomous production of CA215, which might lead to various effects, such as promoting tumor growth [59,71], tumor invasion, migration, and metastasis [72,73]. It can also diminish and evade the activity of immune cells [74,75] and inhibit apoptosis [59,76], all of which can significantly impact the clinical presentation of the disease. This inconsistency can affect TNM staging, prognosis, treatment response, tumor differentiation, and the manifestation of paraneoplastic symptoms, among other factors [77]. Moreover, CA215 retains its effector functions through subdivision Fc, which can activate the neutrophil Fc-gamma-RS receptor [78]. This receptor is an alternative pathway for NETosis initiation [61], as proposed in Figure 3. The production of NETs might be dependent on serum levels of CA215, and so it might influence the outcome of cancer therapy.

## 8. NETs Assessment

Qualitative and quantitative methods were employed by researchers for evaluating the production of NETs in in vivo, in vitro or ex vivo settings, offering insights into their role in inflammatory and other pathological conditions [79,80,81].

### 8.1. Quantitative Assays

The quantitative measurement of NETs provides precise values or concentrations, useful for monitoring their production in various circumstances. The numerical data mainly capture the amount of extracellular DNA, the activity of NETs-associated enzymes such as neutrophil elastase and myeloperoxidase, or even the number of neutrophils that undergo NETosis, typically utilizing serum, full-blood samples, or a culture suspension medium [82,83,84,85]. As for the downsides of using these methods, they may lack the provision of direct evidence regarding various processes within the TME.

*DNA quantification* can directly measure NET formation since they are primarily composed of chromatin fibers [86]. For this purpose, fluorometric assays and quantitative polymerase chain reaction (qPCR) are mostly employed. Fluorometric assays use DNA-intercalating dyes such as Sytox Green, which fluoresce when binding with DNA and allow for the quantity of extracellular DNA to be determined. qPCR targets specific NET genes or repetitive DNA sequences to amplify and quantify extracellular DNA. However, it is essential to ensure that NET-derived DNA is accurately differentiated from other sources of DNA [82,87,88,89,90]. Specific primers for NET-associated DNA sequences include histone-bound DNA fragments, DNA-NE complex, and MPO, which are unique to NETs. This method allows for a precise quantification of NETs [82,87,88,89,90].

*Enzymatic assays*: NETs are often linked to enzymatic activity, such as neutrophil elastase or myeloperoxidase. Therefore, assessing their activity can serve as an indirect measurement of neutrophil extracellular traps. These assays are functional in nature, involving the linkage of the tested molecule to extracellular DNA, which degrades a substrate and alters the final measured absorbance of the main biological product. The enzymatic activity is primarily measured using spectrophotometry [91,92,93,94].

*Flow Cytometry*: This method uses specific markers to evaluate NET formation at the single-cell level. The most commonly used markers for distinguishing between NET-forming and non-NET-forming cells are citrullinated histones (H3, H4), neutrophil elastase, and myeloperoxidase, which can be identified using designated protocols [79,85,94,95]. Once NETs are formed, fluorescent markers are applied for staining. A combination of DNA-binding dyes (such as Sytox Green or DAPI) and antibodies against specific NET components is commonly utilized. The flow cytometer allows for the differentiation between intact neutrophils, apoptotic cells, and NETs based on their size and fluorescence profile [79,85,94,95].

### 8.2. Qualitative Assays

The qualitative measurement of NETs addresses characteristics such as morphologic features and structural aspects. This methodology most commonly utilizes fresh biopsy tissue or fresh culture cell tissue. The morphology of NET formation can be observed in dynamically using techniques such as immunofluorescence microscopy, electron microscopy, or live-cell imaging. This approach provides better insight into the sequential aspects, architecture, or heterogeneity of NET production. However, it is more challenging to analyze and typically requires a well-trained researcher to fully integrate and corelate the data within the experimental or clinical context [81,90,96,97,98,99]. These assays require a multi-faceted approach.

In *immunofluorescence microscopy*, neutrophils stimulated to form NETs on a slide are fixed and stained with fluorescently labeled antibodies targeting NET-specific proteins, along with DNA-binding dyes. This allows for the visualization of NETs’ structure and components under a fluorescence microscope.

In *electron microscopy*, similarly prepared samples are fixed, dehydrated, and embedded in resin. Thin sections are then cut and stained for contrast, providing high-resolution images of NETs and their interactions with pathogens or cells at the ultrastructural level.

Lastly, for *live cell imaging*, neutrophils are cultured on suitable substrates and stimulated to form NETs in a controlled environment, allowing real-time observation of NET formation and dynamics using time-lapse microscopy.

A combination of methods provides a comprehensive qualitative analysis of NETs, revealing their formation, structure, components, and interactions in various contexts [81,90,96,97,98,99].

### 8.3. Conclusions and Future Perspectives

Research into CA215’s role in predicting epithelioma treatment efficacy opens promising avenues in cancer research. Relevant evidence summarized in this review suggests that CA215’s ability to bind to the Fc gamma receptor of neutrophils and induce NETosis is a key factor in influencing treatment outcomes. The intricate interplay between cancer cells, immune cells, and CA215 highlights the complexity of tumor microenvironments and the potential significance of this biomarker in understanding and improving therapeutic responses.

The importance of considering the immune response in the context of cancer treatment is highlighted by the induction of NETosis by CA215, which suggests a potential mechanism through which the immune system actively participates in reducing the efficacy of anticancer treatment. This insight may have implications for the development of novel therapeutic strategies that leverage tumor microenvironment signaling pathways.

While the correlation between CA215 and NETosis appears promising, it is worth acknowledging the limitations of the current research topic. Understanding NETs’ multifaceted impact on cancer progression is challenging due to their diverse functions in tumor growth and therapy response. The TME’s heterogeneity further complicates studying the varying effects of NETs across different cancer types. The use of diverse assessment methods for NETs, including quantitative and qualitative assays, may lead to inconsistent findings if not interpreted properly. The exact mechanisms by which CA215 influences NETosis and cancer progression remain unclear, requiring further research.

Further studies, such as ex vivo experiments or mass cohort examination, are warranted to elucidate the intricacies of this relationship. Longitudinal studies tracking patients’ responses to treatment in correlation with CA215 levels, as well as in vitro experiments elucidating the molecular pathways involved, could provide a better understanding of the prognostic value of CA215. Furthermore, the potential for developing targeted therapies that modulate CA215-mediated NETosis could represent an innovative approach to augment the therapeutic prognosis. Thus, continued research is necessary to fully understand the intricacies of CA215 and unlock its true prognostic potential.

## Figures and Tables

**Figure 1 diagnostics-14-00328-f001:**
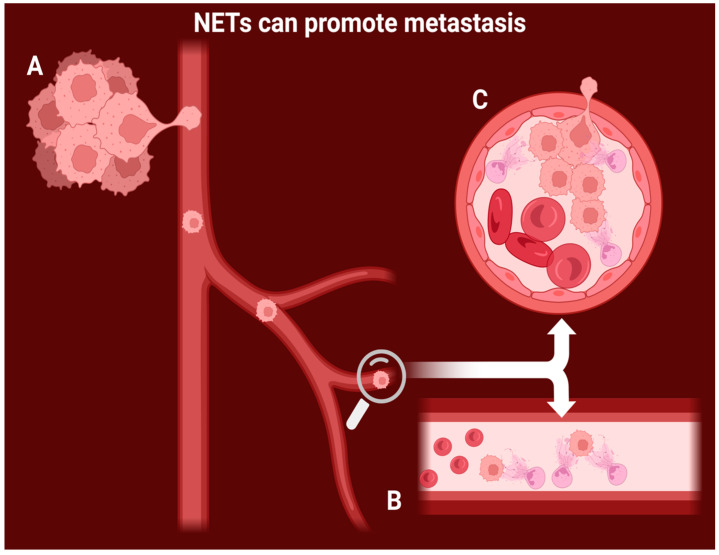
Proposed mechanism for NET-induced metastasis and tumor growth. A: the cancer cells slowly invade the vessel wall and enter the bloodstream. B: intravascular NETs capture the circulating cancer cells and immobilize them. C: NET-mediated tumoral growth leads to vascular occlusion and invasion of new healthy tissue.

**Figure 2 diagnostics-14-00328-f002:**
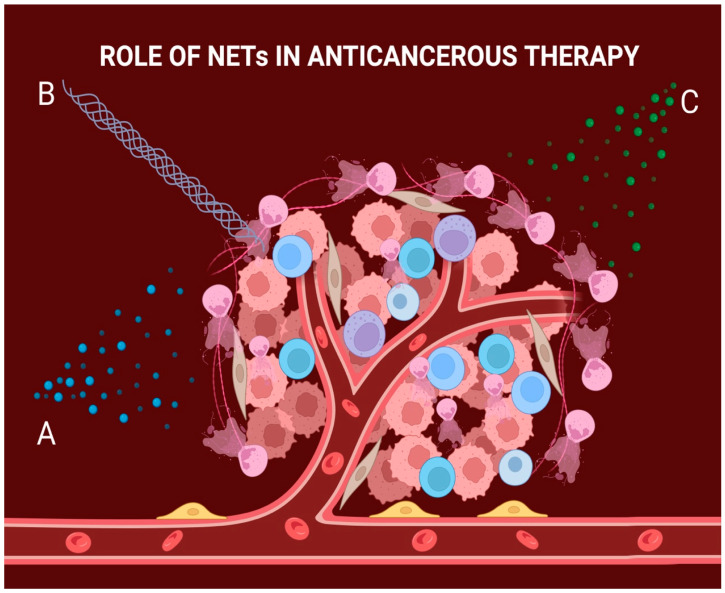
The role of NETs in cancer therapy. A: Cancer cells can internalize the web-like structure which allows them to use it as a strong antioxidant; this causes the chemotherapeutic drug to bind to the structure, resulting in decreased efficacy of the treatment. B: NETs can also organize at the exterior of the tumoral mass, acting as a mechanical barrier to radiation treatment and thus lowering its efficiency. C: NETs can negatively impact the mediation of checkpoint blockade molecules, which in turn lowers the efficiency of immunotherapy.

**Figure 3 diagnostics-14-00328-f003:**
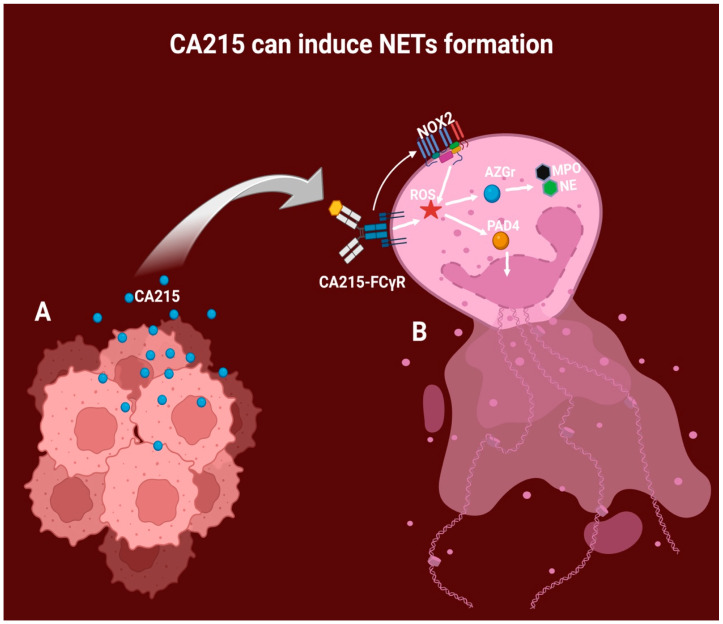
Proposed mechanism of NET formation by CA215 stimulation. A: Epitheliomas can autonomously produce IgG variants, including CA215, which directly impact the tumor microenvironment, including the neutrophils. B: CA215 then binds to the neutrophil receptor FC-gamma, generating intracellular ROS. Additionally, the CA215-FC-gamma complex can stimulate the NOX2 enzymatic system, leading to ROS production. ROS further contributes to the expulsion of DNA via PAD4, generating chromatin decondensation and liberating the content of azurophilic granules, such as MPO and NE.

**Table 1 diagnostics-14-00328-t001:** Different functions of NETs in cancer and cancer therapy.

Role of NETs in Cancer	References
Regulate EndMT	[15,16,17]
Positive effect on tumor progression, invasion, and growth	[18,19,20,21]
Positive effect on angiogenesis	[22,23,24,25]
Provide resistance to chemotherapy	[26,27,28]
Provide resistance to immunotherapy	[31,32]
Provide resistance to radiotherapy	[33,34,35,40,42]

## Data Availability

Not applicable.

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
