# Peer review of "The Role of Neutrophil Extracellular Traps in the Outcome of Malignant Epitheliomas: Significance of CA215 Involvement"

_diagnostics, 2024, doi:10.3390/diagnostics14030328_

Round 1

Reviewer 1 Report

Comments and Suggestions for Authors

This is an interesting article by Himcinschi et al.

In generally, this article is medical hypothesis article, not just review article.

I suggest rewriting this article as hypothesis article, because there are not exist published data that verify the ability of CA215 to induce Neutrophil Extracellular Traps.  

Furthermore, I would like to address a number of suggestions to authors that may improve the manuscript.

General recommendations

Please check all the text for spelling mistakes.

Please correct all references according to journal stile.

References should be described as follows:

Journal Articles:

1.      Author 1; Author 2. Title of the article. Abbreviated Journal Name Year; Volume: page range.

Moreover, in the text, reference numbers should be placed in square brackets [ ],

Introduction

Page 2, lines 47-48

Neutrophils generally play an important role in inflammation, not only in infection.

Considering of main text of this article, that analyzes the role of Neutrophils in cancer, starting with sentence "Neutrophils are the most abundant white blood cells in our immune system and play a crucial role in fighting infections", authors has been confining the role of neutrophils, and are only focused on antimicrobial role of these cells.

I recommend changing this sentence.

The reference [1] "Brinkmann V, Zychlinsky A. Neutrophil extracellular traps: Is immunity the second function of chromatin? J Cell Biol. 404 2012;198(5):773–83." does not reflect to the idea of the text about the discovery of NETs. Please change the reference [1] to "Brinkmann V at al. Science. 2004;303(5663):1532-5. doi: 10.1126/science.1092385."

Page 3 line 99

Is the term "immune infiltrate" correct?

In the literature, most frequently used term is "Tumor infiltrating immune cells (TIICs)" or " Immune cell infiltration."

Page 7 line 284

In the section 6, "CA215 as an Immunoglobulin G variant" authors write that CA215 can bind to the Fc neutrophil receptors and initiate NETosis. Unfortunately, references 61,62 does not confirm that CA215 is able to induce the NETs formation.

The section 8, "Methods for NETs assessment" is not reflect to the title of the article "CA215 induced Neutrophil Extracellular Traps Can Impact the Outcome of Malignant Epitheliomas: a Review".

Moreover, all methods for the assessment of NET formation are well documented in the literature, for example the article by Stoimenou et al. entitled "Methods for the Assessment of NET Formation: From Neutrophil Biology to Translational Research".

Author Response

Dear Reviewer,

We extend our sincere gratitude for the time you invested in reviewing our manuscript and for providing highly constructive comments. Please find below our responses to your insightful comments and suggestions.

Warm regards,

Dr. Adelina Vlad

In generally, this article is medical hypothesis article, not just review article.

I suggest rewriting this article as hypothesis article, because there are not exist published data that verify the ability of CA215 to induce Neutrophil Extracellular Traps.  

We appreciate your suggestion to reframe the article as a hypothesis article. However, we would like to emphasize that our primary intention is to present a comprehensive review paper rather than a hypothesis article. While we acknowledge the absence of published data verifying the ability of CA215 to induce Neutrophil Extracellular Traps (NETs), we believe that the arguments presented in the article hold promise for stimulating further research in this area.

Please consider the following points as supportive of our article being categorized as a review paper:

  1. Comprehensive Analysis of Existing Literature: The document extensively analyses and synthesizes current research on NETs in cancer, including their formation, function, and impact on cancer progression and therapy resistance. The reformulated title of the manuscript, "The Role of Neutrophil Extracellular Traps in the Outcome of Malignant Epitheliomas: Significance of CA215 Involvement," directs the focus toward understanding the implication of NETs in malignant epitheliomas.
  2. Integration of Related Research: The document integrates research on the tumor microenvironment, the role of neutrophils, and cancer antigen 215 (CA215), creating a broader context. This integration aligns with the objectives of a review article to provide a holistic understanding of a topic.
  3. Discussion of Theoretical Foundations: The document discusses the theoretical underpinnings of NETs in cancer, including the potential role of CA215, based on established scientific principles and related research findings. This theoretical exploration is typical of review articles that aim to connect different strands of research.
  4. Identification of Research Gaps: The document identifies gaps in current research, such as the unverified role of CA215 in inducing NETs. Highlighting these gaps is a key aspect of review articles, as they aim to set the stage for future research directions.
  5. Purpose and Audience: The document serves an educational purpose, aiming to inform readers about the current state of research on NETs in cancer. This aligns with the objective of a review article to update the scientific community and interested readers on a specific topic.

Furthermore, I would like to address a number of suggestions to authors that may improve the manuscript.

General recommendations

Please check all the text for spelling mistakes.

Thank you for bringing this to our attention, we have carefully reviewed the entire text and addressed the spelling mistakes to ensure accuracy.

Please correct all references according to journal stile.

We appreciate your diligence in ensuring the quality of our work, all references were carefully corrected.

Moreover, in the text, reference numbers should be placed in square brackets [ ],

The square brackets were added accordingly.

Introduction

Page 2, lines 47-48

Neutrophils generally play an important role in inflammation, not only in infection.

Considering of main text of this article, that analyzes the role of Neutrophils in cancer, starting with sentence “Neutrophils are the most abundant white blood cells in our immune system and play a crucial role in fighting infections”, authors has been confining the role of neutrophils, and are only focused on antimicrobial role of these cells.

I recommend changing this sentence.

Thank you for your observation. The sentence has been modified following the reviewer’s comments (lines 47-48).

The reference [1] “Brinkmann V, Zychlinsky A. Neutrophil extracellular traps: Is immunity the second function of chromatin? J Cell Biol. 404 2012;198(5):773–83.” Does not reflect to the idea of the text about the discovery of NETs. Please change the reference [1] to “Brinkmann V at al. Science. 2004;303(5663):1532-5. Doi: 10.1126/science.1092385.”

The reference has been replaced, thank you for your valuable suggestion.

Page 3 line 99

Is the term “immune infiltrate” correct?

In the literature, most frequently used term is “Tumor infiltrating immune cells (TIICs)” or “ Immune cell infiltration.”

We appreciate your valuable input in guiding our choice. We have chosen “IMMUNE INFILTRATE” as it provides functional arguments regarding the role of immune cells within the tumor microenvironment (TME), a term frequently encountered in relevant literature (references 6, 9, 10, 25 in the manuscript). The term “TIICs” tends to focus more on the infiltration process and the subsequent localization of immune cells within the tumor, with less emphasis on their functional role post-infiltration.

Page 7 line 284

In the section 6, “CA215 as an Immunoglobulin G variant” authors write that CA215 can bind to the Fc neutrophil receptors and initiate NETosis. Unfortunately, references 61,62 does not confirm that CA215 is able to induce the NETs formation.

We acknowledge your concern about the absence of explicit confirmation in references 61 and 62 regarding CA215's ability to induce NETs formation. While it is accurate that there are no direct data linking these two aspects, both references suggest the possibility of a receptor-effector binding between a cell with an intact Fc-gamma-RS receptor and CA-215, which generally possesses an intact effector part similar to a normal IgG1,4. Our intention in this section was to establish a correlation between the studies, with a particular emphasis on the noteworthy work of Vanessa Granger and Silvie Collet-Martin (reference 62), known for their contributions to netosis. It's important to note that the paragraph focuses specifically on CA-215's intact Fc subdivision, as elucidated by Lee et al. in their study (reference 61), rather than addressing CA-215 as a whole molecule. To enhance clarity, we have revised expressions such as "can induce" and "can bind" to keywords like "may" (page 8, lines 300, 301). We hope this adjustment addresses your concerns and improves the precision of our statements.

The section 8, "Methods for NETs assessment" is not reflect to the title of the article "CA215 induced Neutrophil Extracellular Traps Can Impact the Outcome of Malignant Epitheliomas: a Review".

We agree that the current title doesn't entirely match the focus of the "Methods for NETs Assessment" section. To address this, we propose a refinement to the title to better reflect the content and purpose of the article: "The Role of Neutrophil Extracellular Traps in the Outcome of Malignant Epitheliomas: Significance of CA215 Involvement". While "Methods for NETs Assessment" is an important section, it's more of a background or supporting topic, and therefore, it is not included in the title.

Moreover, all methods for the assessment of NET formation are well documented in the literature, for example the article by Stoimenou et al. entitled "Methods for the Assessment of NET Formation: From Neutrophil Biology to Translational Research".

Thank you for your valuable feedback. Although other authors have addressed the subject of NETs assessment, we consider that a brief presentation of NETs determination methods adds significant value to our review, by grounding it in the practical aspects of research, enhancing its educational merit, and contributing to a more nuanced understanding of the topic. By effectively bridging the gap between theoretical knowledge and practical research application, we believe our review becomes a more comprehensive and practical resource for readers.

Reviewer 2 Report

Comments and Suggestions for Authors

The manuscript is well-structured, clearly written, and contributes valuable insights into the intricate interplay between NETs and the tumor microenvironment. I recommend its publication in this journal

Author Response

The manuscript is well-structured, clearly written, and contributes valuable insights into the intricate interplay between NETs and the tumor microenvironment. I recommend its publication in this journal

Dear Reviewer,

Thank you for dedicating time to evaluate our paper and for providing a positive assessment of its overall purpose and significance.

Sincerely,

Dr. Adelina Vlad

Reviewer 3 Report

Comments and Suggestions for Authors

The manuscript provides a comprehensive exploration of neutrophil extracellular traps (NETs) and their intricate involvement in the tumor microenvironment (TME), particularly in the context of cancer development and therapy. With a solid foundation and numerous scientific references, the manuscript elucidates the mechanisms of NETosis and its multifaceted impact on tumor progression, invasion, and resistance to various cancer treatments, including chemotherapy, immunotherapy, and radiotherapy. Notably, it extends beyond NETs to highlight the role of CA215, a pan-cancer marker, in influencing treatment outcomes.

The organization of the manuscript into clear sections enhances readability, but certain issues, may pose challenges for non-expert readers:

1.       Certain concepts and ideas are repeated throughout the manuscript, which may contribute to redundancy. Consider revising to avoid unnecessary repetition and maintain conciseness. 

2.       The section on "Methods for NETs assessment" provides an overview of quantitative and qualitative assays but lacks detailed explanations or examples. Providing more concrete examples or specific methodologies would enhance the practical understanding of these assessment methods. 

3.       While the manuscript discusses various studies and findings related to NETs, there could be more clarity in presenting results. Clearly distinguishing between findings from different studies and summarizing key results in a concise manner could improve comprehension. 

4.       There are instances where terminology is used inconsistently. For example, the manuscript refers to both "NETosis" and "neutrophil extracellular traps (NETs)" without a clear distinction. Maintaining consistency in terminology is essential for clarity. 

5.       Some sections, especially in the latter part of the manuscript, lack subheadings, making it challenging for readers to identify specific topics within these sections. Subheadings can aid in organizing content and guiding readers through the text. 

6.       The manuscript does not explicitly discuss the limitations of the studies or methodologies discussed. Addressing the limitations would provide a more balanced view and help readers interpret the findings critically. 

7.       Some sentences are intricate and may require careful reading to fully comprehend. Consider revising for clarity, ensuring that each sentence conveys its intended meaning with precision. 

8.       There are minor grammatical errors and typos throughout the manuscript. A thorough proofreading session would be beneficial to address these issues and enhance the overall professionalism of the manuscript. 

9.       When presenting study findings, providing comparative analyses between different studies or experimental conditions could strengthen the manuscript. This would enable readers to discern patterns or differences more easily.

Comments on the Quality of English Language

The quality of English language is generally satisfactory, with only minor grammatical issues that could be addressed.

Author Response

Dear Reviewer,

We extend our sincere gratitude for the time you invested in reviewing our manuscript and for providing highly constructive comments. Please find below our responses to your insightful comments and suggestions.

Kind regards,

Dr. Adelina Vlad

The organization of the manuscript into clear sections enhances readability, but certain issues may pose challenges for non-expert readers:

  1. Certain concepts and ideas are repeated throughout the manuscript, which may contribute to redundancy. Consider revising to avoid unnecessary repetition and maintain conciseness. 

We appreciate your feedback regarding potential redundancy in certain concepts throughout the manuscript. To address this concern, we have made revisions as follows: lines 112-113, 195-198. Initially, our focus was on key concepts, including NETs, malignant epitheliomas, CA215, and cancer therapy resistance, employing logical cascades to explain our findings. To mitigate redundancy, we have streamlined the discussion, particularly in complex situations such as the proposed CA215-induced NETosis model.

  1. The section on "Methods for NETs assessment" provides an overview of quantitative and qualitative assays but lacks detailed explanations or examples. Providing more concrete examples or specific methodologies would enhance the practical understanding of these assessment methods. 

Thank you for your recommendation. As suggested, we have expanded on this topic, providing more detailed explanations and examples. Please refer to lines 410-412, 420-423, 429, 435-439, 444, and 448-463 for the enhanced content. We trust that these additions will improve the practical understanding of the assessment methods discussed.

  1. While the manuscript discusses various studies and findings related to NETs, there could be more clarity in presenting results. Clearly distinguishing between findings from different studies and summarizing key results in a concise manner could improve comprehension.

We appreciate your suggestion to enhance clarity in presenting results related to NETs. In response, we have expanded different sections by incorporating additional details to provide more information. This includes summarizing key findings from various studies in a clearer manner. Please refer to lines 237-252, 273-293, 326-333.

  1. There are instances where terminology is used inconsistently. For example, the manuscript refers to both "NETosis" and "neutrophil extracellular traps (NETs)" without a clear distinction. Maintaining consistency in terminology is essential for clarity. 

Thank you for highlighting the inconsistency in terminology. We have made a clear distinction between "NETosis" and "neutrophil extracellular traps (NETs)" in the introduction section (lines 55-57). Additionally, we have revised all terms throughout the manuscript and made necessary changes for consistency.

  1. Some sections, especially in the latter part of the manuscript, lack subheadings, making it challenging for readers to identify specific topics within these sections. Subheadings can aid in organizing content and guiding readers through the text. 

Thank you for your feedback. We have addressed the lack of subheadings in the "Qualitative assays" part. Given the smaller size of the latter sections of the manuscript, we have chosen to mark the subsections only in italic, without additional numbering, to enhance reading fluency.

  1. The manuscript does not explicitly discuss the limitations of the studies or methodologies discussed. Addressing the limitations would provide a more balanced view and help readers interpret the findings critically. 

Thank you for underlining the importance of addressing the limitations in the manuscript. In response to your suggestion, we have incorporated a discussion on the limitations of the studies and methodologies reviewed. For detailed insights, please refer to lines 485-491. We believe that this addition provides a more balanced view and aids readers in interpreting the findings critically.

  1. Some sentences are intricate and may require careful reading to fully comprehend. Consider revising for clarity, ensuring that each sentence conveys its intended meaning with precision

Thank you for your observations. To enhance clarity and facilitate a better understanding of our manuscript, we have made several revisions (please refer to lines 59-60, 83-84, 88-89, 99-100, 136, 150-151, 174, 230, 387-392, 442, 468-469).

  1. There are minor grammatical errors and typos throughout the manuscript. A thorough proofreading session would be beneficial to address these issues and enhance the overall professionalism of the manuscript. 

Thank you for bringing attention to the minor grammatical errors and typos in the manuscript. We have conducted a thorough proofreading session and made the necessary corrections to enhance the overall professionalism of the document.

  1. When presenting study findings, providing comparative analyses between different studies or experimental conditions could strengthen the manuscript. This would enable readers to discern patterns or differences more easily.

Thank you for your valuable suggestion regarding the presentation of study findings. While addressing the third point of your requests, we have expanded on various experimental methods used in different studies to provide intricate details that support NETs research in diverse clinical or experimental contexts. However, due to substantial differences in these contexts and the absence of systematic reviews in the literature offering a detailed comparative analysis of experimental conditions, providing such an analysis ourselves would exceed the scope of our manuscript. We believe that the enhanced discussion of experimental methods adds valuable insights, and we appreciate your understanding of the scope limitations.

Round 2

Reviewer 1 Report

Comments and Suggestions for Authors

The authors have made all corrections based on the input of my recommendations.